# Highly Correlated Recurrence Prognosis in Patients with Metastatic Colorectal Cancer by Synergistic Consideration of Circulating Tumor Cells/Microemboli and Tumor Markers CEA/CA19-9

**DOI:** 10.3390/cells10051149

**Published:** 2021-05-10

**Authors:** Hsueh-Yao Chu, Chih-Yung Yang, Ping-Hao Yeh, Chun-Jieh Hsu, Lu-Wei Chang, Wei-Jen Chan, Chien-Ping Lin, You-You Lyu, Wei-Cheng Wu, Chun-Wei Lee, Jen-Kuei Wu, Jeng-Kai Jiang, Fan-Gang Tseng

**Affiliations:** 1Department of Engineering and System Science, National Tsing Hua University, Hsinchu 30013, Taiwan; kecokoyo@gmail.com (H.-Y.C.); a129187883@gmail.com (P.-H.Y.); chunjiehhsu@gmail.com (C.-J.H.); xxo831230@gmail.com (L.-W.C.); weiren1005@gmail.com (W.-J.C.); evanwu70530@gmail.com (W.-C.W.); waynelee19850217@hotmail.com (C.-W.L.); jkwu@mx.nthu.edu.tw (J.-K.W.); 2Molecular Medicine Research Center, Chang Gung University, Taoyuan 33302, Taiwan; 3Department Education Research, Taipei City Hospital, Taipei 10341, Taiwan; yc3636@hotmail.com; 4Center for General Education, National United University, Miaoli 36003, Taiwan; 5General Education Center, University of Taipei, Taipei 110014, Taiwan; 6Institute of Microbiology and Immunology, National Yang-Ming Chiao-Tung University, Taipei 11221, Taiwan; jainping@gmail.com (C.-P.L.); yylyu14@gmail.com (Y.-Y.L.); 7Nano Science and Technology Program, Taiwan International Graduate Program, Academia Sinica, Taipei 115, Taiwan; 8Biomedical Science and Engineering Center, National Tsing Hua University, No. 101, Sec. 2, Kuang-Fu Rd., Hsinchu 30013, Taiwan; 9Department of Surgery, Division of Colorectal Surgery, Taipei Veterans General Hospital, Taiwan School of Medicine, National Yang-Ming Chiao-Tung University, Taipei 11217, Taiwan; 10Department of Engineering and System Science, Frontier Research Center on Fundamental and Applied Sciences of Matters, National Tsing-Hua University, Hsinchu 30013, Taiwan; 11Research Center for Applied Sciences, Academia Sinica, No. 128, Sec. 2, Academia Rd., Nankang, Taipei 11529, Taiwan

**Keywords:** circulating tumor cell (CTC), CTC clusters, prognostic analysis, colorectal cancer (CRC), liquid biopsy, circulating tumor microemboli (CTM)

## Abstract

Circulation tumor cells (CTCs) play an important role in metastasis and highly correlate with cancer progression; thus, CTCs could be considered as a powerful diagnosis tool. Our previous studies showed that the number of CTCs could be utilized for recurrence prediction in colorectal cancer (CRC); however, the odds ratio was still lower than five. To improve prognosis in CRC patients, we analyzed CTC clusters/microemboli, CTC numbers, and carcinoembryonic antigen (CEA)/carbohydrate antigen 19-9 (CA19-9) levels using a self-assembled cell array (SACA) chip system for recurrence prediction. In CRC patients, the presence of CTC clusters/microemboli may have higher correlation in metastasis when compared to the high number of CTCs. Additionally, when both the number of CTCs and serum CEA levels are high, very high odds ratios of 24.4 and 17.1 are observed in patients at all stages and stage III of CRC, respectively. The high number of CTCs and CTC clusters/microemboli simultaneously suggests the high chance of relapse (odds ratio 8.4). Overall, the characteristic of CTC clusters/microemboli, CEA level, and CTC number have a clinical potential to enhance CRC prognosis.

## 1. Introduction

Cancer was the second leading cause of human deaths worldwide in 2017 [1]. Colorectal cancer (CRC) is ranked as the third most common cancer and the second largest cause of cancer death in Europe and the United States [2,3]. Studies have estimated that the burden of CRC is expected to increase by 60% in the next decade. Moreover, almost half of CRC patients eventually develop metastasis, even with advances in treatment and monitoring [4,5]. Therefore, early diagnosis and recurrence prognosis are the key to preventing death in CRC patients.

Recurrence in CRC patients might be related to the expression of carcinoembryonic antigen (CEA) and carbohydrate cell surface antigen19-9 (CA19-9) [6,7,8]. However, the expression of CEA and CA19-9 might be influenced by several factors, including the liver status, inflammation, and chemotherapy, limiting the precision of recurrence prognosis, especially in advanced-stage cancers [9,10].

Circulating tumor cells (CTCs) are disseminated cells that detach from the primary tumors. They infiltrate the blood vessels and migrate to distant tissues, inducing cancer metastasis [11,12]. The number of CTCs in the blood is associated with disease progression in several types of cancer [13,14,15,16]. Our previous research has shown that the combination of CTC number and CEA levels can improve the precision of cancer prognosis in CRC patients [17]. Several studies have shown that tumor cells would form CTC clusters/microemboli in the stage of metastasis [18,19,20]. Moreover, patients with CTC clusters were more likely to develop metastasis in breast cancer [20,21].

CTC is known to be a type of rare cell, comprising fewer than 100 cells per 10^9^ blood cells in patients [22,23]. The most common approach to detect CTCs relies on antibody-based selection of cells expressing epithelial cell adhesion molecules (EpCAM) on the surface and immunocytochemical identification of cytokeratin-positive and 45 (CD45)-negative nucleated cells [24,25,26]. However, the analysis of CTC clusters is relatively rare in mainstream research due to the fact that these clusters are vulnerable during the pre-treatment process. In addition, it is hard to distinguish CTC clusters using the fluorescence detection systems on the market [27,28,29,30]. To address these issues, we have developed a microfluidics-based cell self-assembling array chip (SACA) [31] and automatic imaging system [32]. The SACA chip can perform multiple rounds of fluid exchange on top of a cellular monolayer, with minimal cell loss. It is possible to perform multiple antibody labeling of CTC for image analysis within 4 h by coupling to an optimized three-dimensional microwell dialysis (3D-microDialysis) chip. This device can identify one CTC per 10^5^ cells.

## 2. Results

### 2.1. The Process for Preparing Clinical Samples

#### 2.1.1. Identification and Detection of CTCs

In brief, samples were preserved by a blood tube with K2EDTA (BD Vacutainer^®^, Plymouth, UK) for both PB and MVB; this stage lasted less than 24 h. Then, red blood cells were removed using Leucosep TM tubes (bio-check LABORATORIES LTD, New Taipei City, Taiwan). In this step, red blood cells were removed through Ficoll–Paque PLUS (GE Healthcare Life Sciences, Taipei, Taiwan) using a concentration gradient. Hoechst (Thermo Fisher Scientific Taiwan Co., Ltd., Taipei, Taiwan) staining was used to mark cell nucleus DNA. CD45 (Beckman Coulter Inc, Brea, CA, USA) was employed to recognize white blood cells. EpCAM (BioMab Inc., Taipei, Taiwan) was used to identify CTC candidates. The details of the process and antibodies were described in a previous work [17]. The SACA chip system was used in CTC and CTC clusters/microemboli enumeration, and the technical details of the SACA chip were described previously [31].

#### 2.1.2. Introduction of SACA Chip

The SACA chip consists of two parts: a 5 μm SU-8 structure made by a lithography process on the glass slide and the acrylic slide with 7 mm wide wells, as shown in the design schematic in Figure 1. Cells are spread on the well bottom by gravity force and capillary fluid force. The side fluid is driven by evaporation force, and cells are restricted by the 5μm gap between well and bottom surface. Each well can contain more than 500,000 cells. The erythrocyte removed samples from a patient’s blood are directly loaded into the well; about 2 SACA chips/15 wells are needed per one blood sample of 7 mL.

#### 2.1.3. Time Evaluation of Each Step for Colorectal Sample

The processes of preparing samples is shown in Figure 2. The process of separating the PBMC from the PB lasted approximately 25 min. Cells were stained by anti-epithelial cell adhesion molecules (EpCAMs) conjugated with fluorescein isothiocyanate (FITC), anti-cluster of differentiation 45 (CD45)-AF594, and Hoechst 33258; the process of staining the cell lasted about 1 h. The cells were settled down on a SACA chip and then became monolayer in 5 min. Finally, all cell images were captured by an automatic imaging system for about 40 min to 80 min. In total, the entire sample process lasted less than 2.5 h.

(The image system protocol is according to Appendix A.)

#### 2.1.4. Automatic Imaging Machine

Our automatic imaging machine was designed in house and manufactured by an external provider (Visionatics Inc., Tainan, Taiwan). The image sensor was chosen from Sony semiconductor products. Both color (IMX183CQJ-J) and monochrome (IMX183CLK-J) products use a high-sensitivity back-illuminated structure with 2.4 μm square unit pixels. The procedure of the machine software is shown in Figure 3. A total of 50 pictures are taken from top to bottom and from left to right for a single well of the SACA chip, and 3 fluorescence images are acquired at the same position before changing to the next place. The system takes 14 min to detect 500,000 cells for 3 fluorescence and one bright view. Overall, the system needs 1 to 2 h to detect CTCs and CTC clusters on a 2 mL blood sample from each patient. 

### 2.2. CTC Phenotyping Analysis in Fluorescence Microscopy

#### 2.2.1. Recovering Rate Analysis in SACA Chip by Spike-In Samples

Fluorescent immunoassay was utilized to detect CTCs in accordance with our previous study [17]. In brief, Hoechst 33258, EpCAM-FITC, and CD45-PECy-7 were used to identify nuclear, EpCAM, and CD45, respectively. HCT 116 CRC cells were spiked into leukocytes using a 1:100,000 ratio and were detected by the SACA chip. The recovering rate was more than 89% (Appendix A). As shown in Figure 4, HCT116 cells were stained in Hoechst 33258 positive/EpCAM-FITC positive/CD45-PECy-7 negative, but leukocytes were stained in Hoechst 33258 positive/EpCAM-FITC negative/CD45-PECy-7 positive. In addition to fluorescence images, the Optical Microscope (OM) images of cells can also be easily acquired using the same SACA imaging system.

As shown in Figure 5, HCT116 cells were also stained in Hoechst 33258 positive/CK-18 positive/CD45-PECy-7 negative and leukocytes were stained in Hoechst 33258 positive/CK-18 negative/CD45-PECy-7 positive. All of them were observed under the same SACA imaging system, which verified that the EPCAM-identified HCT 116 cells can also be stained by CK-18 specific antibody for CRC tumor cells.

#### 2.2.2. Different Types of CTC Detected by the SACA Chip for CRC Patients

CTCs and CTC clusters were detected in blood samples from CRC patients. Figure 6 shows single-cell CTC (Type I) and two types of CTC clusters (Types II and III). Here, the green fluorescence shows the EpCAM^+^ cells, the yellow fluorescence shows CD45^+^ cells, and the blue fluorescence marks the cell nuclei. 

### 2.3. Affirmation in Clinical Samples

#### Patient Characteristics

We collected blood samples from 166 patients. Among them, both peripheral vein blood (PB) and mesenteric vein blood (MVB) samples were collected from 138 patients. The remaining 28 patients provided either PB (25) or MVB (3) samples. As shown in Table 1, samples of PB, MVB, and lymph node metastasis from 166 patients were analyzed dependent on age, sex, and tumor, nodes, and metastasis (TNM) classification stages. The number of PB, MVB, and lymph node metastasis are 163, 141, and 166, respectively. Among all the cases, the percentage of patients earlier than the T2 period was 36%, and 64% of these patients had not developed lymph node metastasis. Nevertheless, 16 patients had developed metastasis in other organs. We analyzed these data by different TMN stages for different clinical meanings.

### 2.4. Clinical Implications of CTCs, CTC Clusters, and CEA in Colorectal Cancer

#### 2.4.1. Results of CTC Detection by the SACA Chip

The number of CTCs from patients with different cancer stages were counted by a SACA chip in the automatic imaging system. Patients were separated into two groups according to their neoadjuvant chemotherapy status. The related statistical data for different cancer stages are shown in Table 2. Figure 7 shows the relationship between CTC counted numbers and the cancer stage of patients. Among them, 65.6% of PB cases were CTC positive (107/163), 70.2% of MVB cases were CTC positive (99/141), and 18.1% of patients had CTC clusters (30/166).

#### 2.4.2. CEA, CA19-9 Value, and CTC Cluster Counts

The increase of CEA and CA 19-9 levels with the increment of disease stage can been observed in Figure 8. However, both tumor markers would be disrupted by liver status, inflammation, and chemotherapy [9,10]. Therefore, a single tumor marker is insufficient to predict disease progression, and more biomarkers are required to improve CRC prognosis.

The percentages of patients with CTC clusters at different cancer stages are shown in Figure 8C. In total, 166 patients with newly diagnosed CRC from a single institute were enrolled, and 30 had detected CTC clusters. (The related cluster count of PB, MVB cases are according to Appendix A.) Three out of 44 patients with stage I CRC had detected CTC clusters, and the ratio was 6.8% (3/44). Similarly, CTC clusters were found in patients with stage II CRC (15.4% (8/52)) and stage III CRC (20.8% (11/53)). It is noticeable that the ratio of patients with CTC clusters reached 50% (8/16) at stage IV CRC. This suggests that CTC clusters may be an important characteristic for CRC prognosis.

### 2.5. Prediction of CRC Recurrence with CTCs, CTC Clusters, CEA, CA19-9, and Their Combination

#### 2.5.1. Recurrence Rate and Odds Ratio Analysis

We calculated the odds ratio (OR) of biomarkers (CTC counts, CEA and CA19-9 concentration, and CTC cluster counts) for recurrence prognosis of CRC. Recurrence rate was defined as (recurrence (+) / total patients), and odds ratio was defined as. The cutoff for the CTC count was 3 cells per 2 mL of blood using the Receiver Operating Characteristic (ROC) curve (Figure 9). The cutoffs with the largest area under the ROC curve (AUC) were chosen. The cutoff of CEA was selected to be 5 ng/mL, and the CA19-9 was selected to be 37 U/mL using clinical statistics from hospital. Because CTC clusters are rare, the presence of CTC clusters is defined as high risk.

#### 2.5.2. Survival Analysis

Kaplan-Meier survival curves on progression-free survival (PFS) are plotted in Figure 10 and Figure 11; the cutoff is similar to the recurrence and odds ratio criteria. We used survival curve to reveal the relation between recurrence and follow-up time.

#### 2.5.3. Prediction of CRC Recurrence at All Cancer Stages

At all cancer stages (163 Peripheral Vein Blood cases, Table 3), the recurrence rate of patients with CTCs > 3 was 32%, but for those with CTCs ≤ 3, it was 6.2% (*p* = 0.0034, OR = 7.1, Figure 10A). Recurrence rates for patients with and without CTC clusters are 45.2% and 6.8%, respectively (*p* < 0.0001, OR = 11.3, Figure 10D). Although CA19-9 showed a high prognosis sensitivity (recurrence rate of CA19-9 > 37 U/mL was 36.4%), it also gave a high false-negative result (recurrence rate of CA19-9 ≤ 37 U/mL was 10.6%); thus the OR value could only reach 4.8. Importantly, CRC prognosis could be improved by combining two or more biomarkers together. For example, the recurrence rate of patients with both CTCs > 3 and CTC clusters was 64.7%, and that of the rest of the patients was 8.2% (*p* < 0.0001, OR = 20.5, Figure 10E). The recurrence rate in patients with CTCs > 3, CEA > 5 ng/mL, and CTC clusters was 75%, but that of the rest of the patients was 11% (*p* < 0.0001, OR = 24.4, Figure 10F). The recurrence rate in patients with CTCs > 3, CEA > 5 ng/mL, CA19-9 > 37 U/mL, and CTC clusters was 100%, but that of the rest of the patients was 12.5% (*p* < 0.0001, OR = N/A, Figure 10G).

#### 2.5.4. Prediction of Recurrence at Stage III CRC

At stage III CRC (51 cases, Table 4), the recurrence rate of patients with CTCs > 3 is 28.6%, but with CTCs ≤ 3 is 13.3% (*p* = 0.0034, OR = 7.1, Figure 11A). The recurrence rate of patients with CEA > 5 ng/mL is 30.8%, but with CEA ≤ 5 ng/mL is 8% (*p* = 0.0369, OR = 5.1, Figure 11B). The recurrence rate of patients with CA19-9 > 37 U/mL is 50%, but CA19-9 ≤ 37 U/mL is 14% (*p* = 0.0048, OR = 6.2, Figure 11C); and the recurrence rate of patients with CTC clusters is 36.4%, but without CTC clusters is 15% (*p* < 0.0001, OR = 11.3, Figure 11D). We found that false-negative results occurred when using CTCs and CTC clusters as prognosis indicators, but CEA and CA19-9 have higher odds ratio at stage III CRC. To improve prognosis, we combined two or more biomarkers together to reach higher prognosis odds ratio, and the results are similar to that analyzed at all stages of CRC.

## 3. Materials and Methods

### 3.1. Sample Resource

The Colorectal cancer samples were sourced from the Taipei Veterans General Hospital for colorectal surgery in Taiwan. The purpose of the trial was to compare the CTC counts with different TMN stages using cell images. The amount of the sample was 133 patients, seen between September 2018 and September 2019, and all the cases were enrolled with a median follow-up of 272 days. The Institutional Review Board (IRB) of Taipei Veterans General Hospital approved the acquisition of the samples, and the case number was IRB-TVPEGH: 2017-11-002ACF. All patients signed the consent for IRB.

### 3.2. Cell Line Culture and Spike-In Testing

The selection of cell lines and the spike-in control method were described in detail in previous work [17]. In brief, HCT116, one type of human colon cancer cell line, was employed in this research; it was injected into white blood cells in order to simulate CTCs in patient blood. HCT116 cells were cultured in a medium containing Roswell Park Memorial Institute (RPMI-1640) (Corning INC., Brooklyn, NY, USA), 10% fetal bovine serum (FBS) (Corning INC., Brooklyn, NY, USA), and 1% penicillin/streptomycin (Pen-Strep) (GIBCO^®^, New York, NY, USA) under standard incubation conditions (37 °C, 5% CO_2_).

### 3.3. Statistical Analysis

The mean ± standard deviation was used to describe the data. Statistical analysis was conducted using Student’s *t*-test. The *p* values less than 0.05 were considered statistically significant. SPSS version 12.0 (IBM, Taipei, Taiwan) and GraphPad Prism 6 (GraphPad Software Inc. San Diego, CA, USA) were used for all statistical analyses. CTC threshold was calculated by the ROC curve method [33]. The time between the day of blood sample collection and the day of local recurrence, distal metastasis, or death was defined as progression-free survival (PFS).

## 4. Discussion

In this research, we demonstrated that using the SACA chip and the auto-imaging system not only distinguish CTCs in the blood of CRC patients but also identify different types of CTC clusters (Figure 4, Figure 5, and Figure 6). We compared CTC numbers in both PB and MVB in blood samples from a total of 133 CRC patients. The results showed that CTC numbers in PB with non-metastatic CRC showed a gradual increase with cancer stage, while the number of CTCs in MVB had no significant correlation (Figure 7). This result is similar to our previous study, showing stability of our SACA system.

CTC clusters/microemboli exist in aggressive cancers, and we speculated that the potential of using clusters/microemboli as the predictor of metastasis would be better than CTC enumeration alone. Our data proved that CTC clusters have more prognostic value than a high number of CTCs. On the other hand, CEA may not be an effective tool for assessing recurrence in metastatic patients; the prognosis efficiency could be greatly increased if regarding CTC counts as an indicator. The combination of CTC and CTC cluster counts can best predict recurrence in patients with non-metastatic CRC, enhancing prognosis dramatically when compared to our previous studies. This research for CRC recurrence prediction only covers one year; long-term observation is still ongoing for CRC patients participating in this study. As a result, we will further expand the number of participating samples for better CRC recurrence prognosis in the future.

## 5. Conclusions

CTC clusters/microemboli play a key role in cancer metastasis and could improve prognosis for CRC patients. Our research successfully utilized a microfluidic-based SACA chip and automatic imaging system to provide meaningful clinical information about CRC patients. The common indicator of CEA may not be an effective predictive tool in metastatic CRC, while the combination of CTCs and CTC clusters/microemboli counts can provide a much superior prediction for recurrence. Moreover, we successfully built up a useful risk stratification tool using information about CTC clusters and demonstrated the feasibility and potential of liquid biopsy in clinical prognosis.

## Figures and Tables

**Figure 1 cells-10-01149-f001:**
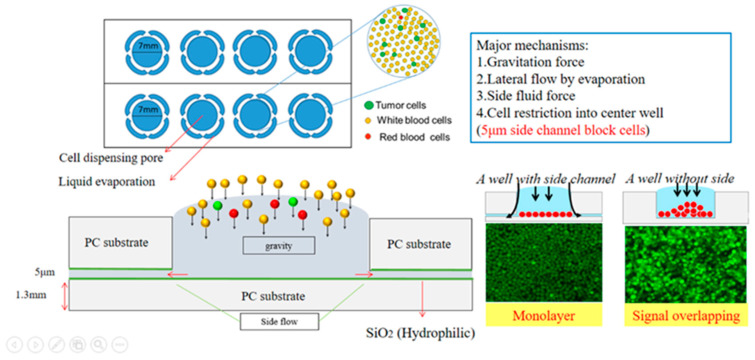
The SACA chip consists of 8 wells with a 5 μm gap on the well bottom for cell restriction during the self-assemble process.

**Figure 2 cells-10-01149-f002:**
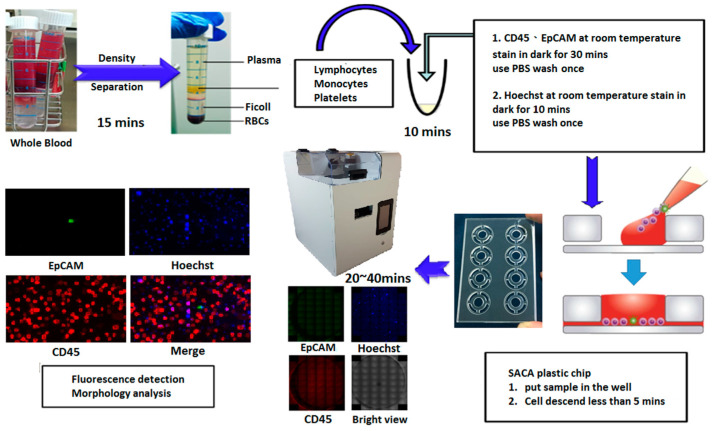
The experimental processes of preparing samples.

**Figure 3 cells-10-01149-f003:**
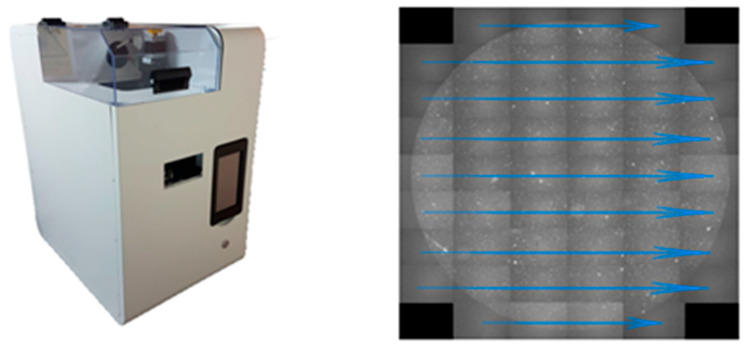
Automatic imaging machine and software procedure.

**Figure 4 cells-10-01149-f004:**
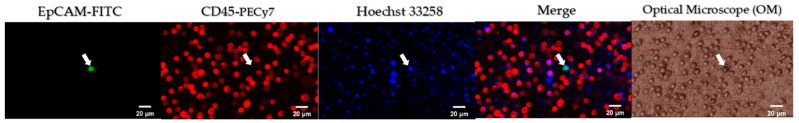
Phenotypic analysis in spike-in samples. The HCT116/leucocyte suspension at 1:100,000 dilution was prepared into a self-assembled cell array (SACA) chip and was stained by anti-epithelial cell adhesion molecules (EpCAMs) conjugated with fluorescein isothiocyanate (FITC), anti-cluster of differentiation 45 (CD45)-AF594, and Hoechst 33258. The targeted cell is marked by an arrow.

**Figure 5 cells-10-01149-f005:**
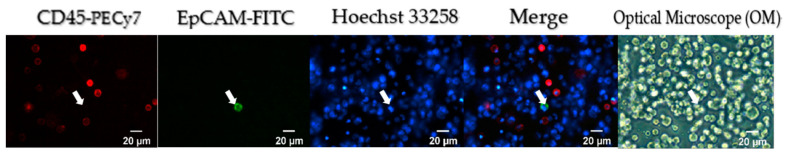
Phenotypic analysis in spike-in samples. The HCT116/leucocyte suspension at 1:100,000 dilution was prepared into a self-assembled cell array (SACA) chip and was stained by CK-18 conjugated with fluorescein isothiocyanate (FITC), anti-cluster of differentiation 45 (CD45)-AF594, and Hoechst 33258. The targeted cell is marked by an arrow.

**Figure 6 cells-10-01149-f006:**
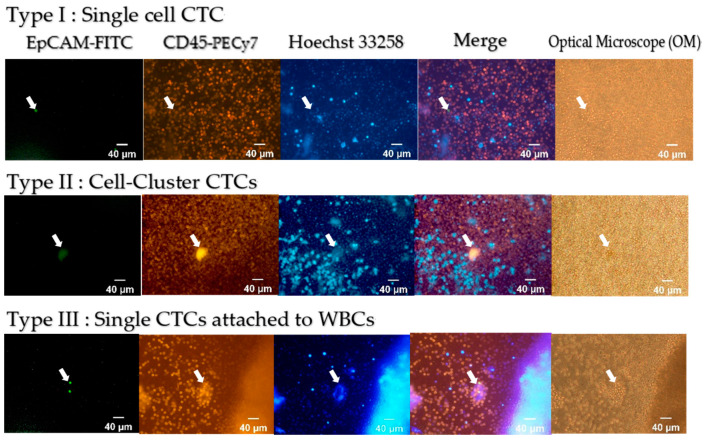
Different kinds of CTC detected by our system.

**Figure 7 cells-10-01149-f007:**
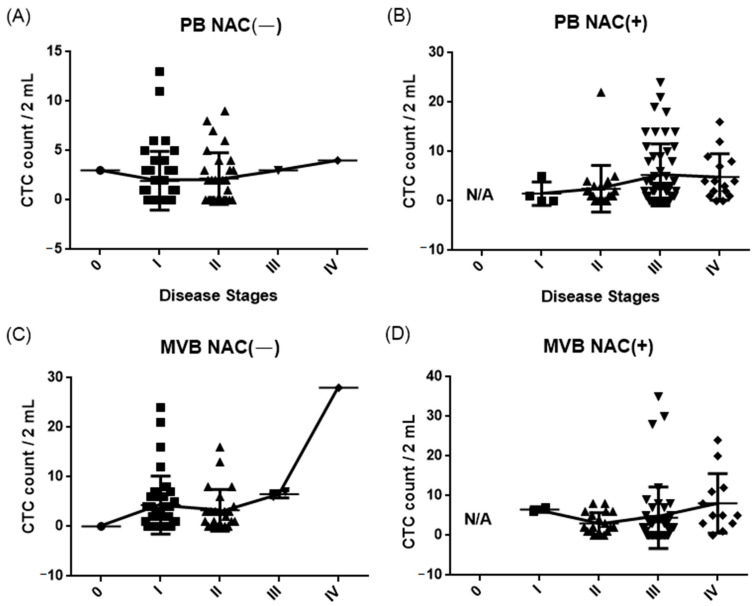
CTC counts for NAC positive or negative patients at different CRC stages from PB (**A**,**B**) and MVB (**C**,**D**) blood.

**Figure 8 cells-10-01149-f008:**
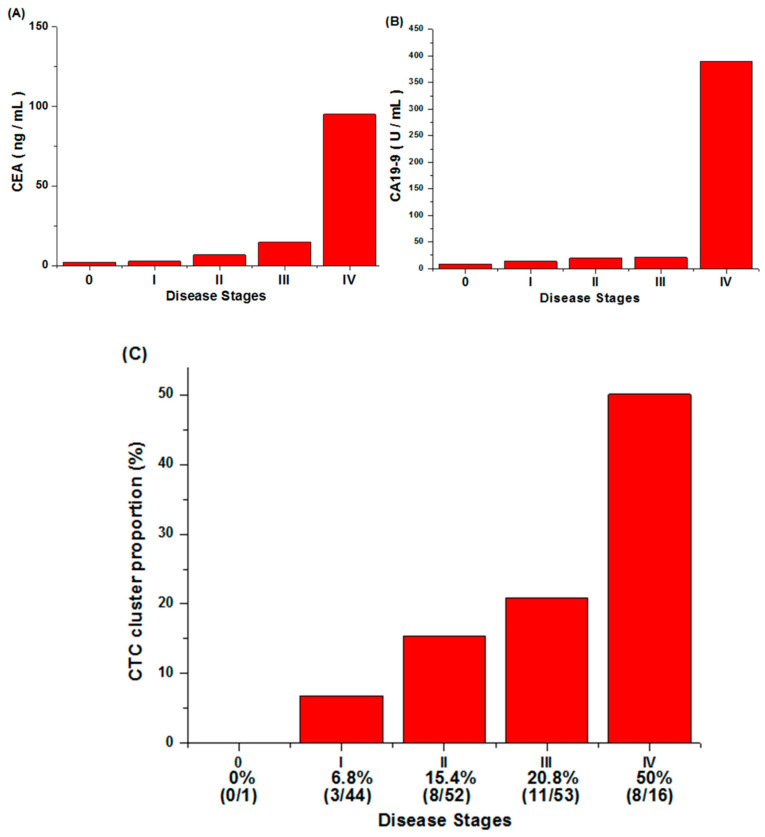
The relationship between CEA/CA 19-9/CTC cluster levels and disease stages. (**A**) The correlation between cancer stages and the carcinoembryonic antigen (CEA). It can be seen that the CEA level has a slight upward trend with the increase in disease stage, and it rises significantly at stage IV. (**B**) The correlation between cancer stages and the carbohydrate antigen 19-9 (CA19-9). The growth trend of CA19-9 in the first three stages is not obvious, but it is significantly increased in stage IV. (**C**) The percentage of patients with CTC clusters at each cancer stage. The CTC cluster proportion increases steadily as the stage increases.

**Figure 9 cells-10-01149-f009:**
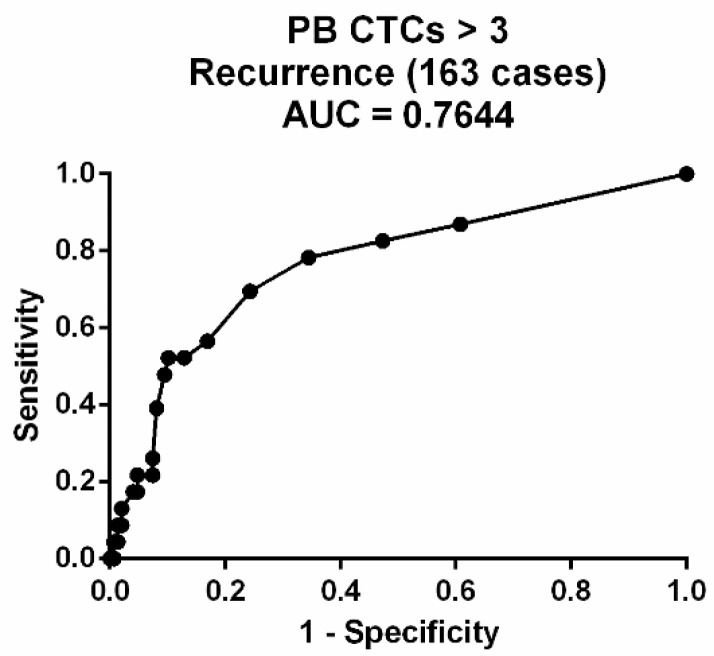
The Receiver Operating Characteristic (ROC) curve of the CTC count from patient blood.

**Figure 10 cells-10-01149-f010:**
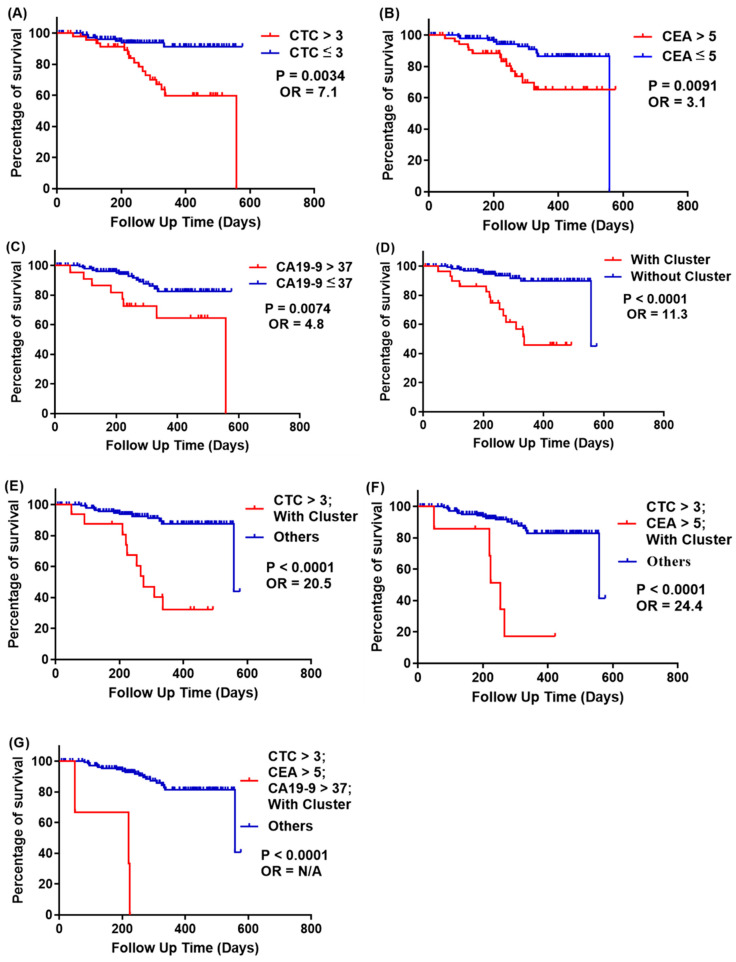
Kaplan-Meier survival curves of progression-free survival at all cancer stages of patients, stratified with CTCs, CEA, CA19-9, and CTC clusters. (**A**) Red line: CTCs > 3; blue line: CTCs ≤ 3; *p* = 0.0034; (**B**) red line: both CEA > 5 ng/mL; blue line: CEA ≤ 5 ng/mL, *p* = 0.0091; (**C**) red line: both CA19-9 > 37 U/mL; blue line: CA19-9 ≤ 37 U/mL, *p* = 0.0074; (**D**) red line: with CTC clusters; blue line: without CTC clusters, *p* < 0.0001; (**E**) red line: CTCs > 3 and with CTC clusters; blue line: others, *p* < 0.0001; (**F**) red line: CTCs > 3, CEA > 5 ng/mL and with CTC clusters; blue line: others, *p* < 0.0001; (**G**) red line: CTCs > 3, CEA > 5 ng/mL, CA19-9 > 37 U/mL, and with CTC clusters; blue line: others, *p* < 0.0001.

**Figure 11 cells-10-01149-f011:**
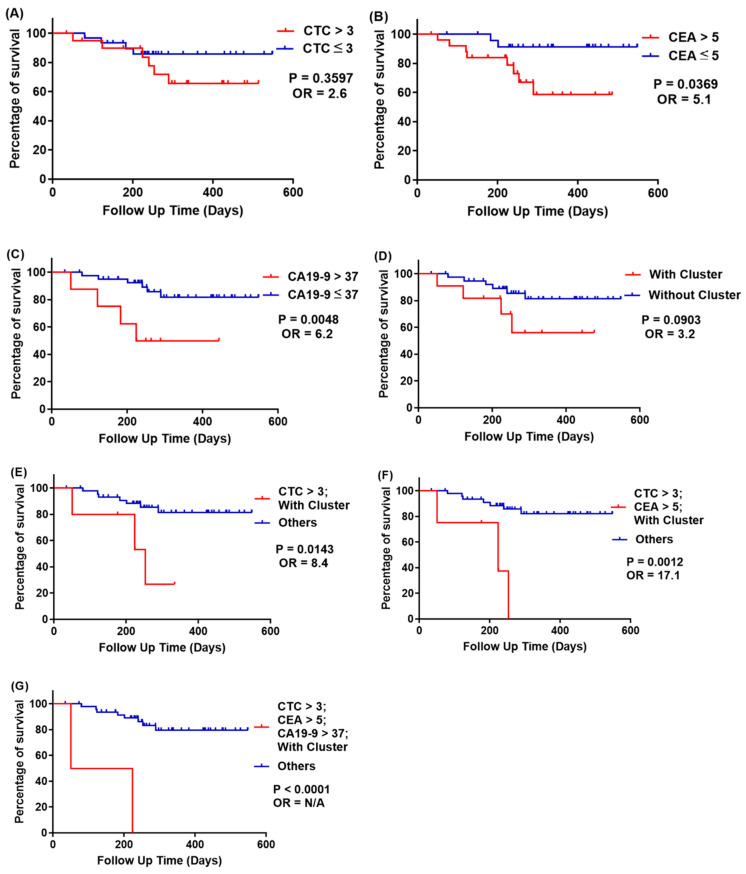
Kaplan-Meier survival curves of progression-free survival of patients at all cancer stages, stratified with CTCs, CEA, CA19-9, and CTC clusters. (**A**) Red line: CTCs > 3; blue line: CTCs ≤ 3; *p* = 0.3597; (**B**) red line: both CEA > 5 ng/mL; blue line: CEA ≤ 5 ng/mL, *p* = 0.0369; (**C**) red line: both CA19-9 > 37 U/mL; blue line: CA19-9 ≤ 37 U/mL, *p* = 0.0048; (**D**) red line: with CTC clusters; blue line: without CTC clusters, *p* = 0.0903; (**E**) red line: CTCs > 3 and with CTC clusters; blue line: others, *p* = 0.0143; (**F**) red line: CTCs > 3, CEA > 5 ng/mL, and with CTC clusters; blue line: others, *p* = 0.0012; (**G**) red line: CTCs > 3, CEA > 5 ng/mL, CA19-9 > 37 U/mL, and with CTC clusters; blue line: others, *p* < 0.0001.

**Table 1 cells-10-01149-t001:** Demographics of patients. Age, sex, and TNM (tumor, nodes, metastasis-classification) stages were analyzed for patients with peripheral blood (PB) and mesenteric vein blood (MVB) samples (**A**,**B**). Lymph node metastasis was analyzed in (**C**). The ratio of carcinoembryonic antigen (CEA) and carbohydrate antigen 19-9 (CA19-9) in the whole assemblage was calculated (**C**).

**(A) Peripheral Vein Blood (PB)**		
Characteristics	Patients (*n*)	Percentage (%)
Age (years)		
<60	58	35.60%
≥60	105	64.40%
Sex		
Male	105	64.40%
Female	58	35.60%
T status		
T0-T2	57	35%
T3-T4	106	65%
**(B) Mesenteric Vein Blood (MVB)**		
Characteristics	Patients (*n*)	Percentage (%)
Age (years)		
<60	52	36.90%
≥60	89	63.10%
Sex	Sex	
Male	92	65.20%
Female	49	34.80%
T status		
T0–T2	50	35.50%
T3–T4	91	64.50%
**(C) All patients**		
Characteristics	Patients (*n*)	Percentage (%)
Lymph Node Metastasis		
Negative	101	60.80%
Positive	65	39.10%
Preoperative Serum CEA		
≤5 ng/mL	55	33.10%
>5 ng/mL	111	66.90%
Preoperative Serum CA19-9		
<37 U/mL	143	86.10%
≥37 U/mL	23	13.90%

TNM: tumor, nodes, metastasis classification.

**Table 2 cells-10-01149-t002:** Number of circulating tumor cells (CTCs) isolated in the colon cancer cases.

**(A) CTC Count in 2 mL PB Blood**													
NAC (−)							NAC (+)						
Disease Stages	No. of Cases	Range	Mean	Median	SD	Mode	Disease Stages	No. of Cases	Range	Mean	Median	SD	Mode
0	1	N/A	3	N/A	N/A	N/A	0	0	N/A	N/A	N/A	N/A	N/A
I	40	0–13	1.95	1	2.94	0	I	4	0–5	1.5	0.5	2.06	0
II	30	0–9	2.17	2	2.27	0	II	21	0–22	2.48	1	4.61	0
III	1	N/A	3	N/A	N/A	N/A	III	50	0–24	5.28	3	6.22	0
IV	1	N/A	4	N/A	N/A	N/A	IV	15	0–16	4.87	4	4.51	4
**(B) CTC Count in 2 mL MVB Blood**													
NAC (−)							NAC (+)						
Disease Stages	No. of Cases	Range	Mean	Median	SD	Mode	Disease Stages	No. of Cases	Range	Mean	Median	SD	Mode
0	1	N/A	0	N/A	N/A	N/A	0	0	N/A	N/A	N/A	N/A	N/A
I	35	0–24	4.31	2	5.76	0	I	2	6–7	6.5	6.5	0.5	N/A
II	25	6–7	3.16	2	4.09	0	II	18	0–8	2.72	2	2.45	2
III	2	0–17	6.5	6.5	0.5	N/A	III	45	0–35	4.44	2	7.69	0
IV	1	N/A	28	N/A	N/A	N/A	IV	12	0–24	8.08	5	7.16	5

NAC—neoadjuvant chemotherapy; SEM—standard error of mean; N/A—not available.

**Table 3 cells-10-01149-t003:** Odds ratios (ORs) of CTCs, CEA, CA19-9, and CTC clusters and their combination to predict recurrence for patients at all cancer stages. N/A: not analyzable due to the absence of recurrent events.

All Cancer Stages (163 Cases)	No. of Cases			
CTCs per 2 mL Blood	Recurrence	No Recurrence	Recurrence Rate	(Odds Ratio, OR)
CTCs > 3	16	34	32%	7.1
CTCs ≤ 3	7	106	6.2%	
CEA > 5	13	41	24%	3.1
CEA ≤ 5	10	99	9.2%	
CA19-9 > 37	8	14	36.4%	4.8
CA19-9 ≤ 37	15	126	10.6%	
With CTC clusters	14	17	45.2%	11.3
Without CTC clusters	9	123	6.8%	
CTCs > 3; CEA > 5	10	12	45.5%	8.2
Others	13	128	9.2%	
CTCs > 3; CA19-9 > 37	4	3	57.1%	9.6
Others	19	137	12.8%	
CTCs > 3; With CTC clusters	11	6	64.7%	20.5
Others	12	134	8.2%	
CEA > 5; CA19-9 > 37	5	7	41.7%	5.3
Others	18	133	11.9%	
CEA > 5; With CTC clusters	7	8	46.7%	7.2
Others	16	132	10.8%	
CA19-9 > 37; With CTC clusters	5	4	55.6%	9.4
Others	18	136	11.7%	
CTCs > 3; CEA > 5; CA19-9 > 37	3	1	75%	20.9
Others	20	139	12.8%	
CTCs > 3; CEA > 5; With CTC clusters	6	2	75%	24.4
Others	17	138	11%	
CTCs > 3; CA19-9 > 37; With CTC clusters	1	1	50%	6.3
Others	22	139	13.7%	
CEA > 5; CA19-9 > 37; With CTC clusters	3	3	50%	6.9
Others	20	137	12.7%	
CTCs > 3; CEA > 5; CA19-9 > 37; With CTC clusters	3	0	100%	N/A
Others	20	140	12.5%	

**Table 4 cells-10-01149-t004:** Odds ratios (ORs) of CTCs, CEA, CA19-9, CTC clusters and their combination to predict recurrence of patients with stage III CRC. N/A: not analyzable due to absence of recurrent.

Stage III CRC (51 Cases)	No. of Cases			
	Recurrence	No Recurrence	Recurrence Rate	(Odds Ratio, OR)
CTCs > 3	6	4	28.6%	2.6
CTCs ≤ 3	15	26	13.3%	
CEA > 5	8	18	30.8%	5.1
CEA ≤ 5	2	23	8%	
CA19-9 > 37	4	4	50%	6.2
CA19-9 ≤ 37	6	37	14%	
With CTC clusters	4	7	36.4%	3.2
Without CTC clusters	6	34	15%	
CTCs > 3; CEA > 5	6	7	46.2%	7.3
Others	4	34	10.5%	
CTCs > 3; CA19-9 > 37	2	0	100%	N/A
Others	8	41	16.3%	
CTCs > 3; With CTC clusters	3	2	60%	8.4
Others	7	39	15.2%	
CEA > 5; CA19-9 > 37	3	3	50%	5.4
Others	7	38	15.6%	
CEA > 5; With CTC clusters	4	5	44.4%	4.8
Others	6	36	14.3%	
CA19-9 > 37; With CTC clusters	3	3	50%	5.4
Others	7	38	15.6%	
CTCs > 3; CEA > 5; CA19-9 > 37	2	0	100%	N/A
Others	8	41	16.3%	
CTCs > 3; CEA > 5; With CTC clusters	3	1	75%	17.1
Others	7	40	14.9%	
CTCs > 3; CA19-9 > 37; With CTC clusters	2	0	100%	N/A
Others	8	41	16.3%	
CEA > 5; CA19-9 > 37; With CTC clusters	3	3	50%	5.4
Others	7	38	15.6%	
CTCs > 3; CEA > 5; CA19-9 > 37; With CTC clusters	2	0	100%	N/A
Others	8	41	16.3%	

## Data Availability

Data for cohort available upon reasonable request.

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
