# Peer review of "Highly Correlated Recurrence Prognosis in Patients with Metastatic Colorectal Cancer by Synergistic Consideration of Circulating Tumor Cells/Microemboli and Tumor Markers CEA/CA19-9"

_cells, 2021, doi:10.3390/cells10051149_

Round 1
Reviewer 1 Report
This manuscript describes the study of colorectal cancer (CRC) recurrence rate based on the scores of CTC, CEA, CA199 and CTC clusters. The authors demonstrated the methodology of analyzing the CTC and CTC clusters with reported SACA chips and an automatic imaging machine. The study involved clinical samples from peripheral vein blood (PB) and inferior mesenteric vein blood (IMVB) from 166 patients for a time span close to 600 days. The percent survival analyses showed that the CTC/CTC cluster diagnosis outperformed the conventional CEA marker. The conclusion is supported by the experimental results. The reviewer recommend the publication on Cells. The supplimentary materails were not found in the link of the manuscript.
Author Response
Point 1: This manuscript describes the study of colorectal cancer (CRC) recurrence rate based on the scores of CTC, CEA, CA199 and CTC clusters. The authors demonstrated the methodology of analyzing the CTC and CTC clusters with reported SACA chips and an automatic imaging machine. The study involved clinical samples from peripheral vein blood (PB) and inferior mesenteric vein blood (IMVB) from 166 patients for a time span close to 600 days. The percent survival analyses showed that the CTC/CTC cluster diagnosis outperformed the conventional CEA marker. The conclusion is supported by the experimental results. The reviewer recommend the publication on Cells. The supplimentary materails were not found in the link of the manuscript.
Response 1: Thanks to the reviewer’s suggestion. This supplementary material has been uploaded. We apologize for this omission.
Reviewer 2 Report
Journal: Cells
Manuscript ID: cells-1008242
Title: Highly Correlated Recurrence Prognosis in Patients with Metastatic Colorectal Cancer by Synergical Consideration of Circulating Tumor Cells/Microemboli and Tumor Markers CEA/CA19-9
Authors: Hsueh-Yao Chu †Chu, Chih-Yung †Yang, Ping-Hao Yeh, Chun-Jieh Hsu Hsu, Chien-Ping Lin, You-You Lyu, Wei-Cheng Wu, Chun-Wei Lee, Jen-Kuei Wu, Jeng-Kai Jiang *, Fan-Gang Tseng *
Work presented is of interest to the readers. Development of the assays utilizing CTC and microemboli to further predict outcomes in the patients diagnosed with cancer or monitor efficacy of the therapies is of importance in the field of translational research in the oncology field. However, statement such as “CTC as one of the key diagnostic tools” should be avoided and removed, as to date these cells were not used in diagnostics, per se.
The major concern in the presented manuscript is lack of appropriate CTC staining for cytokeratins. Classic definition of the CTC; accepted and implemented by the community is: a nucleated cell showing positive staining for cytokeratins and lack of CD45 marker. Why do authors insist on eliminating staining of cells for cytokeratin presence? For example, fluorescence images in Figure S1 do not generate a lot of confidence. It is a disappointment, because a lot of work and effort was clearly put in the collection of these results. This must be addressed by the authors very carefully to convince readers that cells and features in images are truly CTCs and microemboli. More and higher quality images should be presented in the main manuscript and SI to support the conclusions.
The manuscript can be published in Cells only after requested edits are incorporated and clarifications provided.
- The organization of the manuscript is odd and difficult to Please introduce the basic concepts of how the SACA chip works at the very beginning of the manuscript. Elementary characteristics are needed (i.e., what is the surface area of the SACA - how many cells can chip accommodate in a monolayer. How many chips/wells needed per one blood sample of 7 mL?). The reference to previous study does not suffice.
- Presented method should adhere to the standard of immunophenotyping for CTCs. This is especially important that CTC will lose epithelial antigens (i.e., EpCAM) or their expression can be low to begin with. Simple addition of this marker (CK) will make this work really reputable among the peers in the community.
- Please present fluorescent images for microemboli and CTC from patient blood samples, not cell line models in the main manuscript.
- Figure S1 organization is not adequate. Please be consistent with color coding, order of the images showing the same type of stain. What is red panel representing in the middle figure? Looking at this figure, the reviewer is only certain that the cytokeratin must be in the staining panel. The cluster presented in the middle set of images could be platelets aggregates that will also stain for nucleic acids presence and FITC signal is only scatter signal and not real EpCAM (if the green image represent EpCAM, as it is not labeled). This data does not generate a lot of confidence. Controls are needed for these data to stand. And it is a shame, because a lot of work and effort was clearly put in the collection of these results. As of note also, the SI section needs to be polished: Please note EpCAM not Epcam, Cy7 , what is OM in Figure 1 and S1?
- How do you interpret presence of cells observed in the images in Figure 1 that are nucleated but show no CD45 marker nor EpCAM?
- Why the Receiver Operating Characteristic curves are not presented yet mentioned?
- Please use proper language: Patients are not in stages. This is stage of the disease, not patients.
- Please modify the table and Figure 3. Such data follow non-normal distribution, therefore SD are not appropriate to use. Please build box plots to present all data points and show range and medians. Outliers will be shown, as well.
- The same should apply to CEA and CA19-9 values measured in clinical samples. Standard deviations are not used in non-Gaussian distribution of the data.
- Figure 4 and 5 should be combined.
- Please be consistent: odds ratio or probability ratio?
- Again, the structure of the manuscript needs reorganization. Odds ratios in Table 3 should follow Figure 6.
- Figure 8 is redundant with Figure 7.
- Line 145: In figure 4, the concentration of CEA and CA19-9 value increased with stage were showed. Please reword this sentence.
Author Response
Point 1: The organization of the manuscript is odd and difficult to Please introduce the basic concepts of how the SACA chip works at the very beginning of the manuscript. Elementary characteristics are needed (i.e., what is the surface area of the SACA - how many cells can chip accommodate in a monolayer. How many chips/wells needed per one blood sample of 7 mL?). The reference to previous study does not suffice.
Response 1:
Thanks for the suggestion. We didn’t include the detail of SACA chip because it has already been published in another Journal, thus only a citation was placed as a reference in the current manuscript. In order to provide a better background for the readers, in the revised manuscript some essential parts of the SACA chip are included in the introduction part.
Point 2: Presented method should adhere to the standard of immunophenotyping for CTCs. This is especially important that CTC will lose epithelial antigens (i.e., EpCAM) or their expression can be low to begin with. Simple addition of this marker (CK) will make this work really reputable among the peers in the community.
Response 2:
Thanks for your valuable suggestion. It’s true that using EpCAM as the only marker might cause some EMT cells loss due to their missing of epithelial antigen. However the purpose of this research is not to focus on those un-detectable cells by EpCAM anitibody, in instead, we would like to provide a new insight of CTCs and CTMs in prognosis by using the commonly accepted antibodies for better comparison and understanding. In the next several researches, we are planning not only to use multiple immunofluorescent markers such as CK-18 to mark the CTCs, but also use fluorescent dyes such as CD-3, CD-5 to mark immune cells to identify the relationship among immune cells and CTCs.
Point 3: Please present fluorescent images for microemboli and CTC from patient blood samples, not cell line models in the main manuscript.
Response 3:
Different types of CTC have already been presented in the supplement. Figure S1 shows single-cell CTC (Type I), and two types of microemboli (Type II and III).
Point 4: Figure S1 organization is not adequate. Please be consistent with color coding, order of the images showing the same type of stain. What is red panel representing in the middle figure? Looking at this figure, the reviewer is only certain that the cytokeratin must be in the staining panel. The cluster presented in the middle set of images could be platelets aggregates that will also stain for nucleic acids presence and FITC signal is only scatter signal and not real EpCAM (if the green image represent EpCAM, as it is not labeled). This data does not generate a lot of confidence. Controls are needed for these data to stand. And it is a shame, because a lot of work and effort was clearly put in the collection of these results. As of note also, the SI section needs to be polished: Please note EpCAM not Epcam, Cy7, what is OM in Figure 1 and S1?
Response 4:
Tks very much for the detail comments on the supplementary material. The organization and figures are all revised and corrected according to reviewer’s suggestions and commons.
Point 5: How do you interpret presence of cells observed in the images in Figure 1 that are nucleated but show no CD45 marker nor EpCAM?
Response 5:
The group of cells which only present nuclear signal but not others sometimes are found in this research. We considered that those cells might be in the epithelial–mesenchymal transition (EMT) process missing related target surface proteins for our dyes. Those cells were not counted as CTCs to make data more consistent.
Point 6: Why the Receiver Operating Characteristic curves are not presented yet mentioned?
Response 6: Thanks very much for your comments, we have added the Receiver Operating Characteristic curves in the manuscript (Figure 4).
Point 7: Please use proper language: Patients are not in stages. This is stage of the disease, not patients.
Response 7: Thank you for pointing out the mistakes. We have corrected all the related words in the article.
Point 8: Please modify the table and Figure 3. Such data follow non-normal distribution, therefore SD are not appropriate to use. Please build box plots to present all data points and show range and medians. Outliers will be shown, as well.
Response 8:
Thanks for the suggestion. New diagram presenting all data points is now presented in Figure 2.
Point 9: The same should apply to CEA and CA19-9 values measured in clinical samples. Standard deviations are not used in non-Gaussian distribution of the data.
Response 9:
The data presentation style has been corrected accordingly.
Point 10: Figure 4 and 5 should be combined.
Response 10: Thanks for the suggestion. Figure 4 and 5 have already be combined.
Point 11: Please be consistent: odds ratio or probability ratio?
Response 11: Thanks for the suggestion. The related words in the Table have already been corrected.
Point 12: Again, the structure of the manuscript needs reorganization. Odds ratios in Table 3 should follow Figure 6.
Response 12: Thanks for the suggestion. Odds ratios already be added in Figure 6.
Point 13: Figure 8 is redundant with Figure 7.
Response 13: Thanks for the suggestion. All figure orders have already be re-organized.
Point 14: Line 145: In figure 4, the concentration of CEA and CA19-9 value increased with stage were showed. Please reword this sentence.
Response 14:
The sentence has been reworded to “the increase of CEA and CA 19-9 levels with the increment of disease stages can been observed” in page 5, paragraph1, line128.
Reviewer 3 Report
In this study, the authors analyse the contribution of CTC clusters/microemboli together with CTCs numbers, on a self-assembled cell array (SACA) chip system, as well as include the evaluation of tumor CEA/CA19-9 levels for the recurrence prognosis of CRC patients
They conclude that CTC clusters/microemboli play a crucial role in CRC prognosis, and the SACA microfluidic – imaging system might be a promising tool for their analysis.
The study should include a higher number of patient's samples as well as a similar number of normal blood samples, to evaluate the real differences between CRC and normal.
In addition in several of the reported graphs (Fig 3, Fig 4) the SD is very high and data should be added (likely increasing the samples) to better clarify the results.
In addition, the comparison with additional CRC tumor markers (not only CEA/CA19-9) should be included.
Author Response
Point 1: In this study, the authors analyses the contribution of CTC clusters/microemboli together with CTCs numbers, on a self-assembled cell array (SACA) chip system, as well as include the evaluation of tumor CEA/CA19-9 levels for the recurrence prognosis of CRC patients. They conclude that CTC clusters/microemboli play a crucial role in CRC prognosis, and the SACA microfluidic – imaging system might be a promising tool for their analysis. The study should include a higher number of patient's samples as well as a similar number of normal blood samples, to evaluate the real differences between CRC and normal.
Response 1: Thanks to the reviewer’s suggestion. Cellsearch system is currently the gold standard for CTC detection.1-3 In our manuscript, 2.94 CTC were detected in stage I CRC patient per 2ml of blood, equal 10.29 CTC in 7ml blood. The value has exceeded the Cellsearch definition healthy person cut-off value. Of course, we will include healthy people in the future to make the study more complete.
Point 2: In addition in several of the reported graphs (Fig 3, Fig 4) the SD is very high and data should be added (likely increasing the samples) to better clarify the results.
Response 2: we have added the data to better clarify the results in the figure 3 and figure 4. Thank you.
Point 3: In addition, the comparison with additional CRC tumor markers (not only CEA/CA19-9) should be included.
Response 3: We appreciate the reviewer's suggestion. CEA and CA19-9 is the most widely used tumor marker for CRC, that's why we analyzed the relationship between CTCs and two traditional serum tumor markers. Moreover, the high odds ratios have proven the use of CEA and CA19-9 is enough to predict the recurrence prognosis of CRC patients. As you mentioned, we will include more clinical biomarkers to improve the value of CTC clinical prediction in the future.
- Cohen, S. J.; Punt, C.; Iannotti, N.; Saidman, B. H.; Sabbath, K. D.; Gabrail, N. Y.; Picus, J.; Morse, M.; Mitchell, E.; Miller, M. C., Relationship of circulating tumor cells to tumor response, progression-free survival, and overall survival in patients with metastatic colorectal cancer. Clin Oncol 2008, 26, 3213-3221.
- Sastre, J.; Vidaurreta, M.; Gómez, A.; Rivera, F.; Massutí, B.; López, M. R.; Abad, A.; Gallen, M.; Benavides, M.; Aranda, E., Prognostic value of the combination of circulating tumor cells plus KRAS in patients with metastatic colorectal cancer treated with chemotherapy plus bevacizumab. Clinical colorectal cancer 2013, 12 (4), 280-286.
- Tol, J.; Koopman, M.; Miller, M.; Tibbe, A.; Cats, A.; Creemers, G.; Vos, A.; Nagtegaal, I.; Terstappen, L.; Punt, C., Circulating tumour cells early predict progression-free and overall survival in advanced colorectal cancer patients treated with chemotherapy and targeted agents. Annals of oncology 2010, 21 (5), 1006-1012.
Reviewer 4 Report
This work utilized microfluidic-based SACA chip and automatic imaging system to provide meaningful clinical information to CRC patients and built up useful risk stratification tools through the information of CTC clusters, and demonstrated the feasibility and potential of liquid biopsy in clinical prognosis. There are some comments shown below:
- There are quite a few language problems that require extensive text editing or simplification. For examples:
(a) in Line 191-193: "138 patients were collected both peripheral vein blood (PB) and mesenteric vein blood (MVB) samples. 25 and 3 patients can only obtain either PB or MVB, respectively." --> "both peripheral vein blood (PB) and mesenteric vein blood (MVB) samples were collected from 138 patients. The remaining 28 patients provided either PB (25) or MVB (3) samples."
(b) in Line 196, 197: "the percentage of patients which earlier than T2 period" add "are" after "which"; "and 196 64% of these haven’t" add "patients" after "these".
(c) in Line 208, change "number" to "numbers" in order to be consistent with "were".
(d) in Line 260, "Recurrence rate of patients with 260 CTC clusters is 45.2%, but without CTC clusters is 6.8% (p < 0.0001, OR = 11.3, Figure 10D)" can be simplied as "Recurrence rates of patients with and without CTC clusters are 45.2% and 6.8%, respectively(p < 0.0001, OR = 11.3, Figure 10D)."
(e) in Line 265-266, "For example, the recurrence rate in patients with both CTCs > 3 and CTC 265 clusters is 64.7%, and the rest patients is 8.2%" please change "in" to "of"; add "that of" before "the rest patients".
and so on. - In Table 2(A) of NAC(+), there are two "Mean" columns, please correct them.
- In Figure 8. These figures show histograms of (A)CEA, (B)CA19-9, and (C)CTC cluster proportion of patients in different stages. It reflects the correlation between cancer stages and the corresponding tumor-markers or CTC clusters, but the caption should first describe exactly what the figures directly show. Besides, the label along the x-axis for the sub-plots is not consistent and the label of the y-axis of subplot (C) should be changed to be "Percentage of patients with CTC clusters" (also add "s" to the "patient" in Line 233), please correct.
- In Table 3. How the thresholds of CTCs, CEA, and CA199 (i.e. 3, 5, 37, resp.) are determined? Would the result dramatically change if using different threshold values?
- Is there any reason that you specifically studied the prediction of recurrence at Stage II CRC? Please add at least one sentence to describe the reason at the end of Section 2.5.3 or the beginning of Section 2.5.4.
- In Table 4, the numbers of cases in some rows are too small to conduct reliable statistical analysis. For example, row of "CTCs > 3; CA199 > 37" has only 2 cases with 100% recurrence rate, which is not convincing.
- Please change the y-axis label to "percentage of survival" in Figure 10.
Round 2
Reviewer 3 Report
The study has improved.
Author Response
thanks for reviewer's comments